# Tinnitus Prevalence and Associated Factors among Dental Clinicians in the United Arab Emirates

**DOI:** 10.3390/ijerph20021403

**Published:** 2023-01-12

**Authors:** Mohannad Nassar, Md Sofiqul Islam, Stancey D’souza, Milan Praveen, Mohammad Hani Al Masri, Salvatore Sauro, Ahmed Jamleh

**Affiliations:** 1Department of Preventive and Restorative Dentistry, College of Dental Medicine, University of Sharjah, Sharjah 27272, United Arab Emirates; 2Ras Al Khaimah College of Dental Sciences, Ras Al Khaimah Medical and Health Science University, Ras Al Khaimah 11172, United Arab Emirates; 3Dental Biomaterials and Minimally Invasive Dentistry, Departamento de Odontología, Facultad de Ciencias de la Salud, Universidad CEU-Cardenal Herrera, C/Del Pozo ss/n, Alfara del Patriarca, 46115 Valencia, Spain; 4Department of Therapeutic Dentistry, I. M. Sechenov First Moscow State Medical University, 119146 Moscow, Russia; 5Restorative and Prosthetic Dental Sciences, College of Dentistry, King Saud bin Abdulaziz University for Health Sciences, National Guard Health Affairs, Riyadh 11481, Saudi Arabia; 6King Abdullah International Medical Research Centre, National Guard Health Affairs, Riyadh 11481, Saudi Arabia

**Keywords:** tinnitus, noise, dental clinicians, prevalence, ear protection, bothersome, intermittent, constant, dentist, decibel

## Abstract

Dental clinicians are at increased risk for developing tinnitus due to exposure to high levels of occupational noise. This study aimed to determine tinnitus prevalence and associated factors among dental clinicians. Interviews were conducted with 150 randomly selected dental clinicians using a questionnaire. Noise levels were measured at three points of time at the operating area. Tinnitus was reported in 19.33% of the participants. The average noise level was significantly higher among participants with tinnitus compared to those without the condition. More participants complained of intermittent tinnitus and the rest had the constant type, with the latter being significantly more bothersome. The weekly average time of using high-speed handpieces, suction and electric handpieces, age and experience had a significant effect on the presence of tinnitus. Only 2.7% of the participants reported the use of a hearing protection device. Tinnitus is a common finding among dental clinicians especially those with higher levels of occupational noise and more frequent use of noise-generating equipment. Knowledge of tinnitus prevalence aids in realizing the extent of its impact and making informed decisions. These results call for improved awareness of the negative impact of clinically-generated noise and emphasize the importance of preventive measures and periodic audiometry exams.

## 1. Introduction

Frequent exposure to high levels of noise may lead to otologic problems including tinnitus [1]. Tinnitus is the perception of sound without the presence of an actual external stimulus [2] and is closely associated with long term noise overexposure [3]. Unfortunately, complete remission of tinnitus is a rare event [4]. Moreover, there is no standardized treatment for this condition, and the currently available treatment options fail to fully resolve the problem [5,6]. Tinnitus represents a common and distressing problem which affects quality of life [7] and may cause people to have difficulties in sleeping, processing thoughts and concentrating [8]. There is a deficiency in the thorough comprehension of the worldwide prevalence of tinnitus, which is also accompanied by many variations in the data. A recent multinational study of pan-European tinnitus prevalence reported that more than one in seven adults have the condition [9], while it was approximately one in ten adults among the US population [10]. Thus, this condition generates a remarkable amount of unnecessary healthcare costs and a great economic impact [5,6]. Despite the increased interest in tinnitus research and the improved recognition of this condition which is evident by the number of publications [11], it still falls behind other comparable conditions [6].

Dental clinicians in particular are subjected to excessive occupational noise due to the various instruments used daily in the clinic, such as air turbine handpieces and suction systems [12]. The attention to noise in the dental environment is not recent and complaints of tinnitus have been reported since the 1960s [13]. Comparing the level of noise in dental settings in the early literature to those obtained in recent studies, it is possible to note that there has not been any considerable improvement [12,14]. Despite the common perception that dental clinicians are more vulnerable to tinnitus, only a few studies have evaluated the prevalence of this condition among them [14,15,16,17]. The report by Gullikson in 1978 entitled “Tinnitus and the Dentist” is probably the first study on the prevalence of tinnitus among dentists and it provided the initial alarming link between clinic noise levels and tinnitus. In the Gullikson study, 74 dentists were found to have tinnitus comprising 48.7% of the total number of participants [16]. A study conducted in South Africa showed that tinnitus prevalence was 31.85% [17] while this number was 31% in the Myers et al. study which took place in the USA [14]. There has been little attention paid to tinnitus within the dental community in the United Arab Emirates (UAE). Indeed, only two reports related to this matter exist from the UAE; Al-Ali and Hashim reported that 5% of dental clinicians based in the UAE suffered from hearing problems without further elaboration on the types of problems [18]. Furthermore, in 2013 Elmehdi investigated the health effects of noise on the hearing of staff working at dental clinics in the UAE and reported a tinnitus prevalence of 37% [15].

Knowledge of the prevalence of certain conditions aids in realizing the extent of their impact on the studied subjects and in making informed decisions concerning the implementation of appropriate interventions [9]. Tinnitus is a prospective field of study, and its epidemiology is one of the current research interests in its area [11]. Considering the negative impact of tinnitus on the quality of life of certain populations, and the lack of sufficient studies on its prevalence, this survey assessment was aimed at determining the prevalence of tinnitus and its associated factors among dental clinicians practicing in the UAE.

## 2. Materials and Methods

This cross-sectional study was conducted in the UAE and the study population included dental clinicians working in different Emirates. The study was approved by the RAKMHSU-REC and RAK-REC prior to data collection (Ethical approval number RAKMHSU-REC-113-2019-UG-D). A written consent was obtained after explaining the project to the subjects who accepted to take part in the study. The participants were asked to fill in a questionnaire, which was designed to estimate the prevalence of tinnitus within a specific population. The questionnaire started by requesting demographic information such as gender, age and years of experience as dental clinician, followed by inquiring about information related to the set-up of the dental office as multiple dental clinics or a single dental clinic, the participants’ perceptions of the level of noise at their clinics, the average number of hours per week that the dental clinicians use suction, and high-speed, slow-speed, and electric handpieces, and the use of ear protective devices. The participants were asked if they were experiencing tinnitus; and those who reported having the condition were also asked about the nature of their tinnitus (constant versus intermittent) and how bothersome it was.

The noise level at each participant’s clinic was recorded using a decibel meter (in dBA). The measurement was performed three times while the clinician was performing a dental procedure that required the use of noise generating tools or equipment, and the average measurement was considered for analysis.

Statistical analyses were performed using the statistical program SPSS version 22 (IBM, Chicago, IL, USA) at a 5% significance level. Descriptive statistics were created, and the Chi-square test was used to examine the statistical significance of differences in tinnitus condition between groups. Since the clinic noise level data were not normally distributed as shown by the Shapiro–Wilk test (*p* < 0.05), the Mann–Whitney U test was used to compare the clinic noise levels for participants with tinnitus with those recorded at clinics with participants without tinnitus.

## 3. Results

The demographic outcomes of the participants, presence of tinnitus, clinic set-up, noise level perception, and the use of ear protection devices are shown in Table 1. The total number of participants was 150; males 82 (54.7%), and females 68 (45.3%). Dental clinicians who worked at dental offices with multiple dental clinics were twice the number of those who worked at single clinics. More than half of the participants considered the environment at their workplace as noisy. The use of ear protection devices was reported among only 2.7% of the participants. Tinnitus was reported in 29 of the participants which translates into a prevalence of 19.3%.

Table 2 shows factors associated with tinnitus such as age, experience, noise level perception, the use of suction, high speed handpieces and electric handpieces; all these factors were significantly associated with tinnitus (*p* < 0.05). Gender, clinic set-up, and use of slow speed handpieces showed no significant differences between participants with and without tinnitus (*p* > 0.05). It is noteworthy to mention that the prevalence of tinnitus was 16.18% among female participants and 21.05% for male counterparts. Interestingly, participants with an age lower than 30 years showed the highest prevalence of tinnitus, followed by those who were older than 50.

The average noise level at the clinics among all participants was 70.60 ± 7.53 dBA. The average level of noise at clinics of participants with tinnitus (76.12 ± 5.90 dBA) was significantly higher than that recorded at clinics of participants without tinnitus (69.25 ± 7.29 dBA) (*p* < 0.001) (Table 3).

Out of the 29 participants with tinnitus, 18 reported an intermittent type of tinnitus while 11 complained of a constant tinnitus. Only 7 participants out of the 29 with tinnitus categorized the condition as highly bothersome (Table 4). There was a statistical difference between the nature of the tinnitus and the level of bother (*p* = 0.035); among those with the constant type, 45.5% reported a highly bothersome condition while this percentage was 11.1% in participants with intermittent tinnitus (Table 5). However, there was no statistically significant difference within gender in relation to the type of tinnitus (*p* = 0.892) or the level of bother it caused (*p* = 0.840).

## 4. Discussion

It is agreed that the environment in dental clinics is conducive to hearing problems including tinnitus [14]. Loud noises at dental clinics are typically generated through several instruments and devices used daily for several dental procedures; noise level higher than 90 dBA has been reported previously in the literature [19]. Long or frequent exposure to noises above 85 decibels may cause acoustic problems in specific working populations [20]. Tinnitus is considered as a health burden that is increasing in incidence over time and has a negative impact on quality of life [7]. Tinnitus is common within the general population [9,10], and its incidence is expected to be even higher among dental clinicians. There is a worldwide lack of published data focusing on dental clinicians [14], and this study represents the first report on the prevalence of tinnitus solely among dental clinicians in the UAE.

It was observed in our study that approximately 45% of the participants were females and 55% were males. These percentages are not a reflection of the general population demographics in the UAE in which the ratio of males to females is 3:1 [21]. However, the study participants’ ratio (1.2:1) was only slightly different from the gender distribution of UAE dental clinicians reported by Al-Ali and Hashim in 2012, which was 1.5:1 [18]. We attribute the difference to be the result of feminization of dentistry as the proportion of women in this field has been rising steadily [22].

The 19.3% prevalence of tinnitus in this study is in the middle range found in the literature among adult members of the public, at 8%–30%. The deviation from the lower end of the aforementioned range is expected, since our survey was limited to dental clinicians, who were expected to have a higher risk of hearing and acoustic issues [23]. The deviation in our result from the upper end of the aforementioned range is speculated to be the result of limiting the samples to older adults in the studies of the upper range [24]. The prevalence of tinnitus in dental communities in the USA and the UAE was previously reported as 31% and 37%, respectively [14,15]. The differences could have been attributed to different methodologies employed, as well as survey questions and targeted participants. For instance, the authors of the study conducted in the UAE in 2013 were not focused only on dentists as they included administrative staff and dental hygienists, assistants, and technicians, with only a total number of 72 dentists. Noise levels generated by laboratory equipment are known to be excessively high and beyond the permissible limits [25], and hence the inclusion of dental technicians must have amplified the overall prevalence of tinnitus in the study of Elmehdi in 2013 [15].

One of the occupational hazards in the dental profession is the noise generated by different sources such as handpieces and suction systems [26]. Long-term exposure to such noises in dental settings is conducive to tinnitus [27]. The average noise level in the current study is in line with several reports which found levels higher than 65 dBA [19,28]. According to our findings, there are significant differences between participants with tinnitus and those without in terms of actual noise level at the clinics, noise level perception, and the amount of time spent using suction, high-speed handpieces and electric handpieces. The vast majority (93%) of participants with tinnitus used electric handpieces in a range of 0–10 h per week, which reflects that such individuals are depending on other types of handpieces for their clinical work. More than half (65.5%) of the participants with tinnitus had a weekly use of high-speed handpieces for a period of time higher than 15 h. The use of suction devices for more than 15 h per week was also predominant (93%) for participants with tinnitus. Electric handpieces may offer the advantage of being less noisy than air-driven systems, as air-driven handpieces can generate noise up to 94 dBA [29]. Suction-generated noise has been reported to reach levels up to 72 dBA and the unpleasant sound produced is a source of irritation for many [30].

The prevalence of tinnitus among females in our study was 16.18%, while it was 21.95% among male participants with no significant difference. The evidence for a gender difference in the prevalence of tinnitus is equivocal [31]. Several studies found higher prevalence in males [32,33]. There has been no clear understanding of the reason, but there is an assumption that hearing problems in men are more common due to exposure of men to more damaging noise over a lifetime [34] rather than sex-based related differences [31]. Furthermore, hormonal variations between males and females have been reported as a possible factor in the increased incidence among men [34], since estrogens are known to have a protective effect on hearing [35]. It was previously reported that there is an abrupt increase in the occurrence of tinnitus in males around middle age. However, this gender difference was expected to decrease later in life when females start showing hearing loss problems [33]. Despite the slightly greater prevalence in males, females are reported as more likely to be bothered by tinnitus and this was in part attributed to the higher propensity in males to understate the symptoms of illness [32]. However, in our study there was no statistically significant difference by gender in relation to the level of bother. Regardless of gender, bothersome tinnitus is known to negatively impact the quality of life including sleep and thus leads to anxiety and depression [10].

Since there is no treatment that effectively eliminates tinnitus and treatment options are usually directed to ameliorating the negative impact on the quality of life, appropriate preventive measures become more pivotal [6,9]. Ear protection is advocated at a noise level above 85 dBA, and it is mandatory at 95 dBA [29]. Only 2.7% of the participants in the present study reported using hearing protection devices. Unfortunately, very few studies have investigated the use of hearing protection devices among dentists. It is quite disappointing to learn that the use of hearing protection devices is rare within the dental community, despite the recommendation by the American Dental Association council on the importance of noise attenuation through the use of ear plugs, which has been in place since 1974 [36]. Previously reported data on the use of ear protection devices among dental clinicians showed slightly higher percentages than the data obtained in the present study and were around 6.25% and 4.3% in the USA and Brazil, respectively. However, the studies conducted in these two countries included a very small number of participants [37,38]. Lack of knowledge about hearing protection and the desire to learn more was reported among Flemish dentists. Thus, improving awareness of hearing problems, periodic audiological evaluations and adherence to personal protection have been sought [39]. However, an encouraging finding in the Theodoroff and Folmer study was the more common use of hearing protection device by students as compared to dental clinicians [38]. Indeed, the University of Leuven has introduced classes on hearing problems and their prevention into the dental curriculum [39]. Recently, Saliba et al. [40] reported that the majority of dental students (93.8%) were aware that dentists are at increased risk of hearing problems. However, 77.7% knew about hearing protectors and but 3.7% used them. It is essential to focus on dental students in order to develop enduring healthy attitudes and beliefs among future dental clinicians about hearing protection during their dental career [41]. A further crucial preventive measure is to properly maintain and periodically replace dental instruments [42] and this calls for future studies to explore the adherence of dental practitioners to these recommendations.

In the current study, tinnitus was intermittent in the majority (62.1%) of participants, while the rest of the population sample reported a constant type of tinnitus. Both types of tinnitus are considered as a chronic condition [8]. Unfortunately, there is not a well-established definition of intermittent tinnitus, and this condition is poorly described and with great variations in the population [43]. In the Skarżyński et al. study [44], tinnitus was continuous in 66.3% of the patients, while the rest had the intermittent condition. Henry et al. [8] reported on 62 veteran participants with either constant or intermittent type tinnitus, with the latter representing only 37% of the studied sample. Recently, Koops et al. [43] concluded that constant tinnitus was more prevalent than intermittent tinnitus (90% versus 10%, respectively). Meanwhile, other studies found a higher prevalence of intermittent over the constant type. Oiticica and Bittar reported that 68% of patients with tinnitus complained of the intermittent type and this percentage was 75.9% in the Sogebi study [45,46]. Potentially different mechanisms underlying the two conditions were reported in the literature with the possibility of more severe physiological impact encountered with the constant condition, and this might explain the higher percentage of participants with highly bothersome tinnitus among those with the constant type (45.5%) than those with the intermittent type (11.1%). It was speculated that after several years intermittent tinnitus might develop into a constant type for some individuals [43].

Surprisingly, in the current study, the age range with the highest prevalence of tinnitus was 20–31 years, followed by the age range 51–60 and 41–50, 31.4%, 25.9%, and 25.6%, respectively. Tinnitus was least prevalent (2%) in the age group 31–40 years. The high prevalence of tinnitus among young dental clinicians is probably not attributable to occupation-related noise, but rather due to life-style noises. Recently, the exposure of young people to loud music has dramatically increased subsequent to technological innovations with regard to personal smart technology devices such as music players and mobile phones, and the ability to stream and listen to music through these devices [47]. Furthermore, risky patterns of using the earphones and music listening devices have been previously identified [48] and associated with tinnitus [49]. The increased rates of tinnitus in young adults could also be ascribed to leisure noise and higher exposure to loud sounds in social events [7,50,51,52,53]. The reason for the low prevalence of tinnitus among the age group 31–40 years in our study is not clear. However, most of the cases of tinnitus reported globally are above the age of 40 with a peak prevalence above the age of 60 [52,54]. Stohler et al. [7] reported that 80% of tinnitus cases were diagnosed above the age of 40. In the current investigation, only two participants had a total experience of 30 years or more, and one of them reported tinnitus; thus, due to the very limited number of participants in this experience category, caution needs to be exercised in interpreting the high prevalence of tinnitus among this group.

## 5. Conclusions

Although this study has revealed relevant data concerning tinnitus among dental clinicians, some limitations were recognized such as the limited sample size and not including the history of tinnitus and ear trauma or information about lifestyle habits that could have contributed to the presence of tinnitus. The results from the present study confirm the high prevalence of tinnitus among dental clinicians and highlight an alarming risk within the younger generation of dental clinicians. Participants with tinnitus had higher noise levels and noise perception at their clinics compared to participants who did not have this condition. Although not significant, more male dental clinicians suffered from tinnitus than their female counterparts and more participants suffered from intermittent tinnitus. The latter caused lower levels of bother compared to the constant type. The more frequent use of certain tools and equipment had a significant effect on experiencing tinnitus. Ear protection was practiced by only a minor number of dental clinicians. Further research should probably aim at promoting dental clinicians’ awareness of the link between occupational noise and ear-related problems and focusing on the importance of preventive measures such as noise protection equipment gear and periodic audiometry exams.

## Figures and Tables

**Table 1 ijerph-20-01403-t001:** Baseline characteristics.

Variable	Category	N	%
Gender	Male	82	54.7
	Female	68	45.3
Age	≤30	36	24
	31–40	35	23.3
	41–50	51	34.0
	>50	28	18.7
Experience	≤10	56	37.3
	11–20	63	42.0
	21–30	29	19.3
	>30	2	1.3
Clinic setup	Multiple clinics	100	66.7
	Single clinic	50	33.3
Noise level perception	Very noisy	28	18.7
	Somewhat noisy	56	37.3
	Somewhat quiet	52	34.7
	Very quiet	14	9.3
Presence of tinnitus	Yes	29	19.3
	No	121	80.7
Use of ear protection device	Yes	4	2.7
	No	146	97.3

**Table 2 ijerph-20-01403-t002:** Factors associated with tinnitus (* use in hours per week).

Variable	Category	Has Tinnitus		*p* Value (Chi-Square Test)
		N	%	
Gender	Male	18	21.95	0.373
	Female	11	16.18	
Age	≤30	11	30.56	0.023
	31–40	1	2.86	
	41–50	10	19.61	
	>50	7	25.00	
Experience	≤10	12	21.43	0.005
	11–20	5	7.94	
	21–30	11	37.93	
	>30	1	50.00	
Clinic set-up	Multiple clinics	19	19	0.884
	Single clinic	10	20	
Noise level perception	Very noisy	13	46.43	<0.001
	Somewhat noisy	15	26.79	
	Somewhat quiet	1	1.92	
	Very quiet	0	0	
Suction	0–5 *	0	0	<0.001
	6–10	1	4	
	11–15	1	3.85	
	16–20	7	18.42	
	21–25	14	56	
	>25	6	19.35	
High-speed handpiece (Air-driven)	0–5 *	1	5.26	0.050
	6–10	5	15.15	
	11–15	4	10.81	
	16–20	14	34.15	
	21–25	4	28.57	
	>25	1	16.67	
Low-speed handpiece (Air-driven)	0–5 *	12	15.79	0.500
	6–10	16	24.62	
	11–15	1	14.29	
	16–20	0	0	
	21–25	0	0	
	>25	0	0	
Electric handpiece	0–5 *	18	15.52	0.009
	6–10	9	29.03	
	11–15	0	0	
	16–20	0	0	
	21–25	2	100	
	>25	0	0	

**Table 3 ijerph-20-01403-t003:** Noise level (decibel measurement dBA) in the clinics.

	Mean	Median	Q1–Q3	*p* Value (Mann–Whitney Test)
Of participants with tinnitus (n = 29)	76.12	77.20	71.25–81.10	<0.001
Of participants without tinnitus (n = 121)	69.25	69.50	65.35–74.20	
Total (n = 150)	70.60	71.2	67.10–75.50	

**Table 4 ijerph-20-01403-t004:** Nature of tinnitus and level of bother it caused in relation to gender.

Variable	Category	Gender			*p* Value (Chi-Square Test)
		Male	Female	Total (n)	
Nature of tinnitus	Constant	7	4	11	0.892
	Intermittent	11	7	18	
Level of bother caused	Highly	5	2	7	0.840
	Somewhat	7	5	12	

**Table 5 ijerph-20-01403-t005:** Level of tinnitus bother caused in relation to tinnitus type.

Level of Bother Caused	Nature of Tinnitus		*p* Value (Chi-Square Test)
	Constant (n)	Intermittent (n)	
Highly	5	2	0.035
Somewhat	5	7	
Minimally	1	9	

## Data Availability

The data presented in this study are available on request from the corresponding author (M.N.).

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
