# Peer review of "Tinnitus Prevalence and Associated Factors among Dental Clinicians in the United Arab Emirates"

_ijerph, 2023, doi:10.3390/ijerph20021403_

Round 1

Reviewer 1 Report

The authors of this survey assessed the self-reported prevalence of tinnitus among dentists in the Arab Emirates. They stratified their findings by age and sex, and they also identified risk factors for tinnitus related to the job of dental clinicians, mainly the noise level at the workplace. They found a rather high prevalence of self-reported tinnitus of about 20% which calls for preventive action in the view of the authors of the study.

The Methods part would profit from adding some information on how many dentists have been contacted and on the response rate. Is there any evidence that the survey responders were likely representative of all dentists form AE, or is there a possibility that dental clinicians who are affected by the problem were much more likely to participate, which would bias the findings towards a higher tinnitus prevalence?

Also in the Methods, the authors state at the bottom of page 2 that those with tinnitus had to tell the ‘level of bothersome’. How was this done? Was there a visual analog scale (VAS) or what other methodology was used?

The discussion is exceptionally long. It is nevertheless informative, as the authors provide a lot of information on various aspects. This is based on a thorough literature review. Nevertheless, the discussion could maybe be shortened a bit to not lose the reader.

The authors of this survey assessed the self-reported prevalence of tinnitus among dentists in the Arab Emirates. They stratified their findings by age and sex, and they also identified risk factors for tinnitus related to the job of dental clinicians, mainly the noise level at the workplace. They found a rather high prevalence of self-reported tinnitus of about 20% which calls for preventive action in the view of the authors of the study.

The Methods part would profit from adding some information on how many dentists have been contacted and on the response rate. Is there any evidence that the survey responders were likely representative of all dentists form AE, or is there a possibility that dental clinicians who are affected by the problem were much more likely to participate, which would bias the findings towards a higher tinnitus prevalence?

Also in the Methods, the authors state at the bottom of page 2 that those with tinnitus had to tell the ‘level of bothersome’. How was this done? Was there a visual analog scale (VAS) or what other methodology was used?

The discussion is exceptionally long. It is nevertheless informative, as the authors provide a lot of information on various aspects. This is based on a thorough literature review. Nevertheless, the discussion could maybe be shortened a bit to not lose the reader.

Author Response

Reviewer 1

Dear reviewer

On behalf of the co-authors, we would like to extend our sincere appreciation for the time and efforts spent to review the manuscript and it is of a great pleasure to resubmit our manuscript entitled “Tinnitus Prevalence and Associated Factors Among Dental Clinicians in United Arab Emirates”. A list of responses is attached below. We strongly believe that this paper will be of a great interest to the readers of IJERPH and sincerely hope that the manuscript is now suitable for publication in this reputable journal.

Response to comment 1:   

The United Arab Emirates (UAE) consists of 7 Emirates (regions), our cross-sectional study included dental clinicians working in the different regions of UAE. The participants were not contacted by email or telephone to participate in the study, rather the investigators visited random dental clinics to meet with a dental clinician at each location to explain the project and obtain a consent in case of an agreement to participate. The majority of dental clinicians who were approached accepted to take part of the study and only a few of them declined for reasons related mainly to their practice schedule. The investigators had a target of 150 participants in total. The sample size was calculated using an online calculator (Raosoft) with a 95% confidence level and 8% margin of error and the minimum number of samples required for the study was 149. The 19.3% prevalence of tinnitus in this study is in the middle range found in the literature among public adults which is 8%-30%. Thus, we believe that our findings are consistent with the literature of prevalence of tinnitus in individuals with occupational noise. As discussed in the manuscript, the deviation from the lower end of the aforementioned range is expected, since our survey was limited to dental clinicians, who were expected to have a higher risk of earing and acoustic issues due to the noise they are exposed to during their practice. The deviation in our result from the upper end of the aforementioned range is speculated to be the result of limiting the samples to older adults in the studies of the upper range while in our study we included a wider range of age.

Response to comment 2:

The level of bothersome was self-reported, the scale that was used was highly, somewhat and minimal. Unfortunately, tinnitus is not measurable or quantifiable by objective physical recordings and is furthermore not traceable. It is also confined to the individual’s subjective perceptual and emotional experience (Cima 2018). The UK Biobank assessment protocol did not include any tinnitus scale, nor did it measure the loudness of the tinnitus, however the subjective questions included have been deemed reliable in similar studies (Davis 1995; McCormack et al 2014).

Cima RFF. Bothersome tinnitus: Cognitive behavioral perspectives. HNO. 2018;66(5):369-374.

Davis A. Hearing in adults. London: Whurr; 1995.

McCormack A, Edmondson-Jones M, Fortnum H, Dawes P, Middleton H, Munro KJ, Moore DR. The prevalence of tinnitus and the relationship with neuroticism in a middle-aged UK population. J Psychosom Res. 2014;76(1):56-60.

Response to comment 3:

Thank you for the comment on the discussion being informative. We worked on the discussion in a way to cover all studied aspects of the project including but not limited to discussing the gender factor, noise level, type of tinnitus, bothersome level, age factor, use of ear protection, relation to certain tools and equipment used by the participant and hours of use and insights into future studies through elaboration on some gaps in the knowledge which this paper was not able to fill. Despite being quite lengthy, the authors believe that the discussion effectively informs readers on what can be learned from the current tinnitus project and provides context for the results in a way that engages the reader in thinking critically about the tinnitus issue based on an evidence-based interpretation of findings and we try to avoid the discussion being governed solely by objective reporting of information.

Sincerely,

Reviewer 2 Report

The manuscript is reliable, and it gives answers to presented questions. However, the received tinnitus percentages are inside tinnitus prevalence received in general population. Therefore, if the causality between noise, dentistry and tinnitus are sought, this manuscript has problems. In addition, the earlier noise exposure history of the dentists is not described at all. If, the dentists under this study have had exposure of shooting noise, motor sports or domestic noisy hand tools, this could explain the received higher results of tinnitus. I suggest that this dimension would be good to explain better also in this work, and also for example make a table, where tinnitus percentages of general population, dentists in this study, and dentist values received other studies would be presented so that the readers could do their own conclusions.

Perhaps the introduction part could explain better the different proved and suspected causes for the tinnitus, and then it would be easier to find out, that the causes of noise exposure are perhaps not very significant in this study, at least there are also many other causes for the prevalence of tinnitus of the dentists.  

Author Response

Reviewer 2

Dear reviewer

On behalf of the co-authors, we would like to extend our sincere appreciation for the time and efforts spent to review the manuscript and it is of a great pleasure to resubmit our manuscript entitled “Tinnitus Prevalence and Associated Factors Among Dental Clinicians in United Arab Emirates”. A list of responses is attached below. We strongly believe that this paper will be of a great interest to the readers of IJERPH and sincerely hope that the manuscript is now suitable for publication in this reputable journal.

Response to comment: 

Thank you for the comment regarding previous history of exposure to loud sound or noise in the participants. As mentioned in the limitation of the study; although this study has revealed relevant data concerning tinnitus among dental clinicians, some limitations were recognized such as not including the history of tinnitus and ear trauma or information about lifestyle habits that could have contributed to the presence of tinnitus. We also elaborated in the discussion on some aspects that could have resulted in the high prevalence of tinnitus among young dental clinicians which is probably not attributed to occupation-related noise, but it is rather due to life-style noises. Recently, the exposure of young people to loud music has dramatically increased subsequent to technological innovations with regard to personal smart technology devices such as music players and mobile phones, and the ability to stream and listen to music through these devices. Furthermore, risky patterns of using the earphones and music listening devices have been previously identified and associated with tinnitus. The increased rates of tinnitus in young adults could also be ascribed to leisure noise and higher exposure to loud sounds in social events. Thus, we believe the current data are giving several insights into further studies that could focus more on other aspects to pinpoint the causes of tinnitus in a targeted population. 

Sincerely

Reviewer 3 Report

The authors presented a cross-sectional study to investigate tinnitus prevalence and associated factors among dental clinicians. In general, the paper is well written. I have several concerns that need to be addressed.

1.     P7, line 216, “the recommendation by the American Dental Association council on the importance of noise attenuation through the use of ear plugs, which has been in place since 1974”--- Ear protection was practiced by only a minor number of dental clinicians, why? It should be discussed further in the discussion section.

2.     Table 1, This study titled “Tinnitus Prevalence and Associated Factors Among Dental Clinicians”, however. However, other social, economic, and environmental factors that may influence tinnitus were not included in the study.

3.     Table 3, Units of noise level should be added to the table 3. Table 4 &5, “N” should be added in the table.

     4.   Please provide the ethics statement and ethics number in the   method.

Author Response

Reviewer 3

Dear reviewer

On behalf of the co-authors, we would like to extend our sincere appreciation for the time and efforts spent to review the manuscript and it is of a great pleasure to resubmit our manuscript entitled “Tinnitus Prevalence and Associated Factors Among Dental Clinicians in United Arab Emirates”. A list of responses is attached below. We strongly believe that this paper will be of a great interest to the readers of IJERPH and sincerely hope that the manuscript is now suitable for publication in this reputable journal.

Response to comments:

  1. As mentioned in the discussion the lack of adherence to the recommendation might be attributed to lack of awareness about the issue of tinnitus and the lack of classes in dental curricula on the topic of tinnitus. The lack of knowledge about hearing protection and the desire to learn more was reported among dentists in previous studies. Thus, improving the awareness on hearing problems, periodic audiological evaluations and adherence to personal protection have been sought. The University of Leuven has introduced classes on hearing problems and prevention into the dental curriculum as it is essential to focus on dental students in order to develop enduring healthy attitudes and beliefs among future dental clinicians on hearing protection during their dental career. We believe the current data from our project would further highlight the need for classes and continuing professional development lectures on the topic of tinnitus in the region.
  2. Thank you for the comment regarding social, economical, and environmental factors and lifestyle. As mentioned in the limitation of the study; although this study has revealed relevant data concerning tinnitus among dental clinicians, some limitations were recognized such as not including the history of tinnitus and ear trauma or information about lifestyle habits that could have contributed to the presence of tinnitus. We also elaborated in the discussion on some aspects that could have resulted in the high prevalence of tinnitus among young dental clinicians which is probably not attributed to occupation-related noise, but it is rather due to life-style noises. Recently, the exposure of young people to loud music has dramatically increased subsequent to technological innovations with regard to personal smart technology devices such as music players and mobile phones, and the ability to stream and listen to music through these devices. Furthermore, risky patterns of using the earphones and music listening devices have been previously identified and associated with tinnitus. The increased rates of tinnitus in young adults could also be ascribed to leisure noise and higher exposure to loud sounds in social events. Thus, we believe the current data are giving several insights into further studies that could focus more on other aspects to pinpoint the causes of tinnitus in a targeted population. 
  3. The noise unit was added to Table 3 and the letter n was added to Table 4 and Table 5.
  4. The ethical approval statement and number are added to the methodology section. 

Round 2

Reviewer 2 Report

OK